# Robust Cytonuclear Coordination of Transcription in Nascent *Arabidopsis thaliana* Autopolyploids

**DOI:** 10.3390/genes11020134

**Published:** 2020-01-28

**Authors:** Jeremy E. Coate, W. Max Schreyer, David Kum, Jeff J. Doyle

**Affiliations:** 1Department of Biology, Reed College, Portland, OR 97202, USA; wmaxwellsch@gmail.com (W.M.S.); dakum@reed.edu (D.K.); 2School of Integrative Plant Science, Cornell University, Ithaca, NY, 14853, USA; jjd5@cornell.edu

**Keywords:** polyploidy, *Arabidopsis thaliana*, cytonuclear interactions, transcription

## Abstract

Polyploidy is hypothesized to cause dosage imbalances between the nucleus and the other genome-containing organelles (mitochondria and plastids), but the evidence for this is limited. We performed RNA-seq on *Arabidopsis thaliana* diploids and their derived autopolyploids to quantify the degree of inter-genome coordination of transcriptional responses to nuclear whole genome duplication in two different organs (sepals and rosette leaves). We show that nuclear and organellar genomes exhibit highly coordinated responses in both organs. First, organelle genome copy number increased in response to nuclear whole genome duplication (WGD), at least partially compensating for altered nuclear genome dosage. Second, transcriptional output of the different cellular compartments is tuned to maintain diploid-like levels of relative expression among interacting genes. In particular, plastid genes and nuclear genes whose products are plastid-targeted show coordinated down-regulation, such that their expression levels relative to each other remain constant across ploidy levels. Conversely, mitochondrial genes and nuclear genes with mitochondrial targeting show either constant or coordinated up-regulation of expression relative to other nuclear genes. Thus, cytonuclear coordination is robust to changes in nuclear ploidy level, with diploid-like balance in transcript abundances achieved within three generations after nuclear whole genome duplication.

## 1. Introduction

Plant cells must coordinate the activities of three distinct genomes—those of the nucleus, plastids and mitochondria, and failure to do so is associated with disruption of such fundamental metabolic processes as respiration and photosynthesis [1,2,3,4]. Cytoplasmic substitution lines exhibit widespread deleterious effects on fitness traits [3,5,6], indicating that the three genomes co-evolved to optimize their interactions, and that altering these interactions can have negative effects on cellular function [7]. 

Disruptions to cytonuclear interactions has long been thought to be a central challenge faced by polyploids [8,9,10,11,12]. Allopolyploidy, which involves the merger of two or more differentiated nuclear genomes, also brings together novel combinations of nuclear and cytoplasmic genomes, potentially creating cytonuclear incompatibilities. These incompatibilities have been the major focus of research on cytonuclear interactions in polyploids [9,12,13,14,15,16].

Polyploidy also increases nuclear genome content without directly altering the dosage of the cytosolic genomes, potentially causing dosage imbalances [8,9,12]. Because many protein complexes in organelles consist of both organelle-encoded subunits and nuclear-encoded subunits [17], altering relative gene dosage potentially disrupts the assembly and/or function of these chimeric complexes [18]. This could have immediate deleterious consequences in both allo- and autopolyploids (polyploids resulting from genome multiplication in the absence of hybridization). This aspect of cytonuclear interactions has received less attention than the cytonuclear incompatibilities specifically associated with allopolyploidy [8].

Nuclear genes whose products are targeted to organelles are among the first to return to single copy following whole genome duplication (WGD) [19]. This has been hypothesized to be the result of selection against genome dosage-imbalance, suggesting that WGD may deleteriously alter dosage relationships between nuclear and cytoplasmic genomes. But few studies have directly assessed if this is true. Does nuclear genome doubling alter the stoichiometry of nuclear-encoded vs. organelle-encoded subunits of chimeric protein complexes, or is inter-genome coordination capable of compensating for nuclear genome doubling (e.g., via increasing the number of organelles and/or organellar transcription)? 

Oberprieler et al. [8] examined this question for two chimeric complexes of the plastid (the large and small subunits of RuBisCO and the PsbA and PsbO subunits of photosystem II) in diploid, tetraploid and octoploid species of *Leucanthemum*. They showed that although the copy number of plastid-encoded psbA decreased relative to that of the nucleus-encoded psbO with increasing ploidy, the ratio of transcript abundances for the two genes remained constant. This indicates that for these two interacting proteins, the nuclear and plastid genomes are able to coordinate expression levels to compensate for changes in relative gene dosage. Similarly, in the case of RuBisCO, they showed that rbcL expression per gene copy is upregulated in the polyploids, which also suggests some coordination in response to WGD. In contrast to psbA/psbO, however, the increase in rbcL expression did not keep pace with the increase in rbcS expression, indicating that dosage compensation was incomplete. Thus, mechanisms of dosage compensation appear to be acting in both cases, but with variable degrees of effectiveness. This could reflect a higher tolerance for transcriptional dosage imbalances in some complexes vs. others [10].

Aside from the work of Oberprieler et al. [8], we know of no studies that have directly examined the stoichiometric effects of WGD on cytonuclear interactions. Moreover, Oberprieler et al. [8] examined responses in established, natural polyploids, which may be auto- or allopolyploids [20], so the immediate responses to doubling (autopolyploidy) remain unclear. Additionally, their work focused on two chimeric complexes of the plastid, meaning no studies have examined responses in mitochondrial complexes, or extended such analyses to assess genome-wide effects. Thus, much remains to be understood about the extent of inter-genome coordination of gene expression in response to WGD. 

Here, we employed RNA-seq in two different organs (sepals and rosette leaves) of synthetic autotetraploids of *Arabidopsis thaliana* (L.) Heynh, coupled with targeting information from the Cytonuclear Molecular Interaction Reference for Arabidopsis (CyMIRA) database [17], to quantify the degree of inter-genome coordination in response to WGD. By utilizing nascent autopolyploids, we could assess the immediate effects of nuclear genome multiplication, independent of hybridization, on transcriptional coordination between the different genome-containing compartments of the cell. If compensatory mechanisms exist to maintain balanced cytonuclear interactions in the face of altered nuclear genome dosage, we hypothesized that:To compensate for increased nuclear dosage relative to organelle dosage, nuclear genes whose protein products are targeted to organelles will be down-regulated in autopolyploids relative to diploids, and/or organellar genes of the autopolyploids will be up-regulated.To maintain proper stoichiometry with interacting proteins encoded by organelles, nuclear genes whose products are targeted to organelles will exhibit less variation in transcriptional response to genome doubling than do other nuclear genes.

To test these hypotheses, we measured relative expression of nuclear, mitochondrial and plastid genomes. We then used targeting and interaction data from the CyMIRA database to assess transcriptional responses in the different compartments to determine if nuclear genes with organelle-targeting show transcriptional coordination with organellar genes and/or more constrained responses relative to other nuclear genes. Additionally, we quantified absolute dosage responses (transcripts per gene copy) to determine what effect WGD has on transcript stoichiometry.

## 2. Materials and Methods

### 2.1. Plant Material

Colchicine-induced tetraploid and octoploid *Arabidopsis thaliana* accession Col-0 were generated as described [21]. Seeds of these lines as well as their diploid parent were provided by Dr. Dana Robinson and Dr. Adrienne Roeder. C24 and Ws tetraploids were generated in the lab of Dr. Luca Comai as described for other accessions in Pignatta et al. [22]. For each accession, diploid and tetraploid lines were derived from the same colchicine-treated plant. All plants used in the current study were of the 3rd generation post-colchicine treatment. 

Warschau (Wa) diploids were also generated in the lab of Dr. Luca Comai. Tetraploid Wa was crossed with the Tailswap haploid inducer to produce diploid seed as described in Ravi and Chan [23]. Wa plants used in the current study were 3rd generation post-haploid induction. 

Seeds were sown on moist Sunshine Mix #4 growing media (Sungro Horticulture, Agawam, MA, USA), kept at 4 °C for four days, and grown in a growth chamber with approximately 125 µ mol/m^2^/*s* light intensity. Sepals were dissected from Col-0 plants grown under a 24 h photoperiod at 22 °C. Tissue was collected from flowers in the final bud stage (stage 12; [24]), and immediately frozen in liquid nitrogen. Three biological replicates were collected per ploidy level, with 100–200 sepals from 2–4 individuals pooled per replicate. All samples were collected between 1 p.m. and 4 p.m. to minimize the effects of diurnal variation in gene expression. Mature rosette leaf tissue was collected from C24 and Wa plants grown under a 16 h photoperiod with 21/18 °C light/dark temperatures. Tissue was collected between 1–3 hours into the light period from fully expanded rosette leaves at the 10–12 leaf stage.

### 2.2. RNA Extraction and RNA-Seq

Total nucleic acids (DNA and RNA) were extracted from rosette leaves using the AllPrep DNA/RNA Mini kit (Qiagen, Germantown, MD, USA) following the manufacturer’s instructions. Total nucleic acids were extracted from sepals as follows: sepals were pulverized in 200 µL GTC Lysis buffer (Omega Bio-Tek, Norcross, GA, USA) containing 4 µL of β-mercaptoethanol, then passed through a homogenizer column (Omega Bio-Tek) by centrifugation at 15,000× g. The flow through was mixed with one volume of 100% EtOH, and nucleic acids were recovered following the clean and concentrator-5 protocol (Zymo Research) beginning at step 3. Nucleic acids were eluted with 50 ul of nuclease-free water. The resulting total nucleic acids were split into two fractions, and one was treated with Turbo DNase (Invitrogen, Waltham, MA, USA) following the manufacturer’s instructions to yield purified RNA. The other fraction was treated with RNase cocktail enzyme mix (Invitrogen) at room temperature for 10 min to recover purified DNA. RNA and DNA concentrations and yields were measured using Qubit RNA Broad Range and DNA High Sensitivity assays (Life Technologies, Waltham, MA, USA). The size of the total RNA transcriptome (total RNA per unit of DNA) was estimated by the ratio of RNA to DNA.

Purified RNA was spiked with ERCC Spike-in Mix 1 (Ambion, Waltham, MA, USA) in proportion to the DNA to RNA ratio obtained from Qubit assays. For sepals, strand specific RNA-Seq libraries were made from spiked RNA using the NEBNext Ultra Directional RNA library kit for Illumina (New England Biolabs, Ipswich, MA, USA) at the RNA Sequencing Core in the College of Veterinary Medicine at Cornell University. All nine libraries (three ploidy levels × three biological replicates) were multiplexed and sequenced on an Illumina NextSeq 500 (75 nt read lengths) (Illumina, San Diego, CA, USA). For leaves, RNA-seq libraries were generated using the Illumina TruSeq Stranded library prep kits. Libraries were multiplexed with 8–12 samples per lane and 100 bp single end sequences were generated on an Illumina HiSeq 250 at the Cornell Biotechnology Resource Center’s genomics facility. Different library protocols were used for sepals and leaves because we generated leaf libraries in house, whereas sepal libraries were made at a core facility. Because all comparisons were between ploidy levels within an organ (leaves or sepals), the different library protocols should not affect interpretation of results.

### 2.3. RNA-Seq Data Analysis

The analysis of RNA-seq output files was completed using a Linux OS virtual machine accessed from the CyVerse Atmosphere platform (https://atmo.cyverse.org/). The instance generated for this study was based on the DataCarpentry Genomics May2019 v1.8.1 image (https://atmo.cyverse.org/application/images/1699).

Trimming: Trimmomatic 0.39 [25] (http://www.usadellab.org/cms/?page=trimmomatic) was used to prepare files for mapping with the following parameters:java -jar trimmomatic-0.39.jar SE -phred33 **{input filename}**.fastq.gz **{output filename}**.fastq.gz ILLUMINACLIP:**{path to adapter directory}**/TruSeq3-SE.fa:2:30:10 LEADING:3 TRAILING:3 SLIDINGWINDOW:4:15

After trimming, output files were checked for sequence quality using FastQC 0.11.8 (https://www.bioinformatics.babraham.ac.uk/projects/fastqc/). 

Mapping: Once files were trimmed and checked for sequence quality, the sequence reads were mapped to the TAIR10 assembly of the *A. thaliana* genome using HISAT2 2.1.0 (https://ccb.jhu.edu/software/hisat2/index.shtml). The TAIR10 reference genome was downloaded from Phytozome 13 (https://phytozome.jgi.doe.gov/pz/portal.html) and indexed using the HISAT2 -build command:hisat2-build -f Athaliana_447_TAIR10.fa Athaliana_index

The trimmed FASTQ files were mapped to the indexed reference genome using:hisat2 -p 8 -x Athaliana_index -q **{input filename}**.fastq -S **{output filename}**.mapped.sam -t

Quantifying transcription: Read counts per gene were determined from the BAM files using HTSeq [26] with the “intersection non-empty” setting using the Araport11 annotation file (Athaliana_447_Araport11.gene_exons.gff3) downloaded from Phytozome (https://phytozome.jgi.doe.gov/pz/portal.html). Normalized read counts (transcripts per million [TPM]) were calculated by first calculating reads per kilobase (RPK; read count per gene divided by gene length in Kb), summing RPK values for all genes, then dividing individual RPK values by the sum of all RPK values divided by 1 million.

Statistics: TPM ratios (e.g., 4C/2C) were compared among targeting classes or genomes using Kruskal–Wallis tests, followed by Dunn’s multiple comparisons tests with Benjamini–Hochberg adjustments for multiple comparisons. Total read count ratios were compared by one-way ANOVA followed by t-tests with Bonferroni correction for multiple comparisons. Correlations between expression estimates from poly(A)-selected libraries and ribosomal RNA-depleted libraries (see below) were assessed by one-way ANOVA. All statistical tests were implemented in R. Dunn tests were implemented using the FSA package for R (https://github.com/droglenc/FSA). 

### 2.4. Comparison of Poly(A)-Selected RNA-Seq Libraries to Ribosome-Depleted RNA-Seq Libraries

To assess the utility of our polyA-selected RNA-seq libraries for quantifying organellar expression, we compared the leaf organelle expression profiles of our libraries to ribosome-depleted libraries acquired from the SRA (BioProject PRJNA437291 accessions SRR6814473, SRR6814474 and SRR6814480, and BioProject PRJNA380344 [GEO: GSE96994] accessions SRR6156820 and SRR6156821). BioProject PRJNA437291 was a study of the effects of herbivory [27]. The libraries were prepared from equivalent tissues (rosette leaves from pre-flowering plants) collected from plants grown under similar conditions (in soil with a 16-hour photoperiod) as our plants, but rather than using poly(A) selection, they were generated from RNA after ribosomal RNA depletion using the Illumina TruSeq Stranded Total RNA kit with Ribo-zero plant kit, and sequenced on an Illumina HiSeq 2500 to produce SE 50 bp reads. BioProject PRJNA380344 was a study of DNA methylation. As above, plants were grown in soil with a 16-hour photoperiod and libraries were generated from rosette leaves of 1-month-old plants. The libraries were generated from ribosomal depleted RNA (Ribo-zero plant kit; Illumina cat. No. MRZPL1224). Libraries were constructed with the Scriptseq kit (Epicentre, cat. no. SSV21124) and sequenced on an Illumina HiSeq 4000 to generate SE 100bp reads. We selected the libraries generated from untreated, wild type control plants from both projects. Libraries were selected from two different experiments to have independent assessments of the correlations between poly(A)-selected and rRNA-depleted libraries. The raw Fastq files were downloaded from the SRA using the SRA-toolkit, then processed and mapped with the identical pipeline used for our in-house libraries.

## 3. Results

### 3.1. Organelle Copy Number per Nuclear Genome

We performed qPCR to quantify plastid and mitochondrial genome copy number relative to nuclear genome copy number. Using genomic DNA from diploid and tetraploid rosette leaves (extracted from the same plants used for RNA-seq), we amplified plastid-encoded RbcL (ATCG00490) or mitochondria-encoded ribosomal 26S (ATMG00020), and normalized their copy number to the geometric mean of two nuclear genes (lipid-transfer protein [AT2G10940] and protein phosphatase 2A [AT1G69960]). The qPCR data indicate that plastid copy number decreased slightly per nuclear genome (0.76-fold ± 0.18 SE) in tetraploids vs. diploids, whereas mitochondrial copy number remained constant (1.01-fold ± 0.25 SE) (Figure 1). Thus, tetraploids appear to compensate for increased nuclear genome dosage with a comparable increase in mitochondrial genome copy number, and partially compensate with an increase in plastid genome copy number. Note, however, that because the relative copy numbers reported here are based on limited gene sampling (two nuclear genes and two organellar genes), the estimates should be considered approximate. Quantitative PCR with additional genes and/or direct counting by microscopy will help to validate and refine these estimates.

### 3.2. Nuclear Gene Expression by Subcellular Targeting

We calculated normalized expression levels (transcripts per million; TPM) for all genes in the most recent annotation of the Arabidopsis genome and plotted expression levels by subcellular targeting using targeting assignments from the CyMIRA database [17] (http://cymira.colostate.edu/). In all three cytotypes (2C, 4C and 8C) and both organs (sepals and rosette leaves), genes with plastid- and dual-targeted products had the highest median expression, and genes with unknown targeting were expressed at the lowest levels (Figure 2).

To assess transcriptional responses to WGD for the different targeting classes, we calculated ratios of TPMs (tetraploid/diploid, octoploid/diploid and octoploid/tetraploid) for each gene, and plotted the distributions of these ratios by targeting category (Figure 3). Distributions of TPM ratios differed significantly by targeting category (*p* < 0.001; Kruskal–Wallis test). In both organs, the expression of genes whose products have unknown or non-organelle targeting (CyMIRA targeting designation of “other”) increased with WGD (e.g., 4C/2C ratio >1), whereas the expression of genes whose products are targeted to plastids generally decreased (ratio < 1). Genes with mitochondrial targeting exhibited intermediate responses, with the median falling just above 1 in most cases. 

This indicates that the fraction of total nuclear transcription targeted to plastids decreases with increasing ploidy. This finding is also supported when we sum read counts from all nuclear genes encoding plastid-targeted proteins and divide this by the sum of read counts for all protein-coding nuclear genes (plastid-targeted / all nuclear). Expression from nuclear genes with plastid-targeted products became a smaller fraction of total nuclear expression with increasing ploidy (Figure 4A,C). Notably, and conversely, expression from nuclear genes encoding mitochondria-targeted proteins became a larger fraction of total nuclear expression with increasing ploidy in both organs (Figure 4B,D). 

Consequently, as ploidy increases, relative transcriptional output within the nucleus increases for genes with mitochondrial targeting and decreases for genes with plastid targeting. These expression patterns closely reflect genome copy numbers (Figure 1). Namely, the decrease in plastid-targeted expression mirrors the decrease in plastid genomes per nuclear genome, whereas mitochondria-targeted expression reflects the greater increase in mitochondrial genome copy number per cell, at least in rosette leaves (Figure 1).

To see if this restructuring of the nuclear transcriptome reflects coordination with organellar transcription, we next looked to see if WGD caused shifts in the overall level of transcription derived from each of the three genomes in the cell (nuclear, mitochondrial and plastid). Before doing this, however, we first assessed how accurately read counts from our RNA-seq libraries reflect actual expression levels for organellar genes.

### 3.3. Organelle Expression Profiles Are Correlated between Poly(A)-Selected and rRNA-Depleted Libraries

Our RNA-seq libraries were generated from polyadenylated transcripts, but transcripts from organellar protein-coding genes are not always polyadenylated [28], and generally targeted for degradation if they are [29,30]. Thus, read counts from organellar genes may not reflect the relative abundances of the transcripts produced. To assess how accurately read counts correlate with expression levels for organellar genes, we compared expression estimates from our libraries to those of RNA-seq libraries generated from rRNA-depleted rather than poly(A)-selected samples. We obtained rRNA-depleted libraries that were generated from diploid plants grown under similar conditions to our own from the Sequence Read Archive, and processed the raw Fastq files using the same pipeline (see Methods). We then compared normalized expression estimates (TPMs) from rRNA-depleted libraries to those of our diploid libraries. For all three genomes, expression levels were significantly correlated between our poly(A)-selected leaf libraries and rRNA-depleted RNA-seq libraries (Table 1, Figure 5). 

As expected, the main difference between the two library types was that organellar TPMs were higher in rRNA-depleted libraries than in poly(A)-selected libraries (Figure 5). As a result of this relative depletion of organellar transcripts, the poly(A) libraries exhibit higher average TPM values for nuclear genes. The significant correlations between expression profiles, however, suggest that the reduced recovery of organellar transcripts in the poly(A) libraries affected most organellar genes proportionately, and that poly(A)-selected libraries, therefore, accurately report relative expression profiles for organellar genes.

### 3.4. Organellar Gene Expression

To assess if WGD alters the relative levels of transcriptional output from organelles vs. the nucleus, we examined the ratio of organelle expression to nuclear expression. Similar to above (Figure 4), we summed read counts derived from organellar genes and divided by the sum of read counts derived from nuclear genes (Figure 6). In both sepals and leaves, the ratio of plastid transcripts to nuclear transcripts decreased with WGD, though the trend was not significant in either case (Figure 6A,C). This pattern parallels the pattern observed for nuclear genes with plastid targeting (Figure 4A,C). In contrast, the ratio of mitochondrial transcripts to nuclear transcripts increased with WGD (Figure 6B,D), again mimicking the pattern observed for nuclear genes with targeting to the mitochondria. 

We see a similar pattern when we calculate TPM ratios for each genome (Figure 7). Plastid TPMs decrease with increasing ploidy, and mitochondrial TPMs increase in tetraploids relative to diploids in both organs. We note, however, that by this method, octoploid sepals deviate from this general pattern for mitochondrial expression, in contrast to the bulk read count approach shown in Figure 6. This is explained by the fact that the bulk read count method is disproportionately influenced by highly expressed genes, whereas TPM ratios weigh all genes equally. ATMG00020, which encodes the 26S ribosomal RNA, is the most highly expressed gene in the mitochondrial genome (accounting for 38%–64% of all mitochondrial reads per library), and is 1.4-fold more highly expressed in octoploid sepals vs. diploid sepals, accounting for much of the increase shown in Figure 6B. We note, however, that in every comparison and by both methods (Figure 6 and Figure 7), plastid expression decreases with increasing ploidy.

Together, the data suggest that the nucleus and organelles are able to coordinate expression responses to nuclear genome doubling within the first few generations post-WGD (plastid and plastid-targeted nuclear transcription decrease in a coordinated fashion, while mitochondrial and mitochondria-targeted nuclear transcription increase in an at least partially coordinated fashion). If such coordination is in fact the case, we would expect the ratio of organelle-encoded transcription to organelle-targeted nuclear transcription to remain relatively constant across ploidy levels. Indeed, the ratio of plastid-encoded transcription to plastid-targeted nuclear transcription is stable across ploidy levels in both organs (Figure 8). Thus, overall, expression levels of both plastid genes and nuclear genes encoding plastid-targeted proteins decreased but remained constant relative to each other following WGD, suggesting precise transcriptional coordination between the two compartments. 

In contrast, although the expression of mitochondrial genes and nuclear genes encoding mitochondrial proteins both increase to some extent, the ratio of mitochondrial transcripts to mitochondria-targeted nuclear transcripts also increases with ploidy (Figure 9). This general pattern holds even if we exclude ATMG00020. Thus, coordination of transcription between compartments does not appear to be as well conserved across ploidy levels in the case of mitochondria as it is with plastids.

### 3.5. Transcriptional Responses to WGD in Nuclear Genes Encoding Organelle-Targeted Proteins 

Among the nuclear genes with targeting to the plastid or mitochondria or both (4268), 910 encode proteins that are classified in the CyMIRA database as interacting with organellar proteins (they are components of chimeric protein complexes), RNA or DNA [17]. Of these, 154 genes encode proteins that are likely to have direct interaction with organelle-encoded proteins. To the extent that cytonuclear interactions are coordinated at the level of transcription, we predicted that the genes encoding these interacting proteins should show the greatest degree of relative expression level homeostasis across ploidy levels. We, therefore, assessed if nuclear genes encoding organelle-targeted proteins showed more coordinated responses to WGD than do other genes. As shown by the smaller deviations from the 1:1 line in the scatter plots of tetraploid vs. diploid TPMs, genes with targeting to both organelles showed more uniform responses than did genes whose products are targeted elsewhere (CyMIRA targeting designated as “other”; Figure 10A,E and Figure 11A,E). As predicted, nuclear genes whose products interact with organelle-encoded products showed even more constrained responses to WGD than do genes with organelle-targeting overall (Figure 10B,E and Figure 11B,E), and nuclear genes whose products directly interact with organelle-encoded proteins show the greatest constraint (Figure 10C,F and Figure 11C,F). The degree of coordination in transcriptional response to WGD is summarized by the coefficient of variation (CV) of TPM ratios (4C/2C) for the different classes of genes (Figure 10D,H and Figure 11D,H).

We next analyzed transcriptional responses to WGD within chimeric organellar protein complexes (plastid-localized RuBisCO and photosystem II, and mitochondria-localized oxidative phosphorylation complex III [OXPHOS complex III]) (Figure 12A–C). For reference, we also examined responses within an entirely nuclear-encoded mitochondrial protein complex (OXPHOS complex II; Figure 12D). 

In the chimeric complexes, organelle-encoded and nucleus-encoded subunits show similar, highly uniform responses (r > 0.99; tetraploid TPM vs. diploid TPM), comparable to the entirely nuclear-encoded subunits of OXPHOS II. These patterns are consistent with robust, inter-genome coordination. 

### 3.6. Transcriptional Responses per Cell to WGD

The preceding analyses were based on relative expression estimates (TPMs or ratios of total read counts). Such relative measures report the fraction of total transcripts contributed by a given gene (i.e., transcript concentration; [31]), but do not, by themselves, allow inferences about absolute expression levels, such as transcripts per gene copy or transcripts per cell [31,32,33,34]. Total transcript abundance (transcriptome size) generally increases with ploidy [21,31,32,33,35], meaning that the expression of individual genes could decrease in relative expression (e.g., TPM’s, transcript concentrations), as observed for genes with plastid-targeting, while remaining constant or even increasing in absolute expression per gene or per cell. 

To determine how absolute transcript abundance responds to WGD, we employed an exogenous spike-in strategy that enabled us to estimate absolute expression levels (transcripts per gene copy and transcripts per cell) as described previously [33] and reported for sepals in Robinson et al. [21] and for rosette leaves in Song et al. [36]. Briefly, we first co-extracted DNA and RNA from a given organ (sepals or leaves) and quantified the amount of total RNA per unit of DNA. Then, prior to generation of RNA-seq libraries, we added ERCC RNAs to the total plant RNA in proportion to the amount of DNA associated with that amount of RNA in planta. For example, we recovered 19% more DNA per unit of RNA, on average, from diploid C24 leaves than from tetraploid C24 leaves (indicating a slight reduction in total RNA transcriptome size per genome in the tetraploids), so we added 19% more ERCC RNA to diploid C24 total RNA than to the equivalent amount of tetraploid C24 total RNA. We then normalized plant-derived read counts by ERCC read counts to estimate expression per unit of DNA (equivalent to expression per gene copy). 

Figure 13 shows the distributions of transcripts per gene copy for nuclear genes broken down by targeting class. In sepals, expression per gene increased in tetraploids vs. diploids for all targeting classes, indicating a >1:1 dosage effect, on average (Figure 13A). Octoploid sepals, on the other hand, show reduced expression per gene relative to diploids for genes with organelle-targeting (Figure 13B), and relative to tetraploids for all targeting classes (Figure 13C). Similarly, in rosette leaves, the expression per gene decreased with WGD for all targeting classes (Figure 13D). Thus, in most cases, WGD is associated with partial dosage compensation for all targeting classes, but dosage responses to WGD differ subtly by both organ and ploidy level. Nonetheless, the same pattern of relative expression observed at the level of TPMs (Figure 2) persists in which genes with plastid-targeting show lower transcript concentrations (as a result of greater dosage-compensation or weaker dosage effects) than do other nuclear genes.

## 4. Discussion

It has often been speculated that WGD disrupts the coordinated functioning of nuclear and cytoplasmic genomes, both by disrupting co-evolved complexes (particularly in allopolyploids), and by disrupting inter-genome dosage balance. Several lines of evidence support the conclusion that the genes encoding chimeric protein complexes in organelles have co-evolved [33], and that hybridization can produce incompatibilities that disrupt their function [9,13,14,15,16]. However, surprisingly little experimental evidence has been gathered to test if WGD, in the absence of hybridization (autopolyploidy) disrupts dosage balance between the nucleus and organelles. The evidence that changes in genome dosage could be problematic is mostly indirect. For example, nuclear genes whose products are organelle-targeted are among the first to return to single copy following WGD [19], which has been hypothesized to be the result of selection against genome dosage-imbalance.

Here, we show that nuclear and plastid genomes exhibit coordinated dosage responses to WGD in nascent autopolyploids. Specifically, plastid genes and nuclear genes whose products are plastid-targeted exhibit coordinated reductions in relative expression (TPMs). Consequently, the ratio of plastid gene expression to plastid-targeted nuclear gene expression remains constant across ploidy levels. Additionally, nuclear genes with plastid-targeting exhibit more uniform responses to WGD than do other nuclear genes, which is indicative of constraints on relative transcript dosage between the nucleus and plastid. Finally, plastid-encoded and nuclear-encoded subunits of chimeric protein complexes exhibit equivalent responses to WGD. Thus, even though the plastid genomes are not directly duplicated themselves, plastid genes maintain balance in transcript stoichiometry with their nuclear partners.

Together, these observations support the assertion that coordination of transcription between the nucleus and plastid is important, and that such coordination is maintained, even in the initial stages (within the first three generations) after nuclear genome duplication. Thus, there appear to be robust mechanisms in place to maintain balanced transcriptional output between the nuclear and plastid compartments, even in the face of altered nuclear dosage resulting from WGD.

In contrast, the maintenance of transcriptional balance between the nucleus and mitochondria following WGD does not appear to be quite as robust, such that the relative expression of nuclear genes with mitochondria-targeting decreases relative to that of mitochondrial genes following WGD (Figure 8). But even in the case of mitochondria, we see some evidence for parallel upregulation of transcription in both compartments, suggesting that there are mechanisms in place to at least partially buffer imbalances that might otherwise be produced by WGD. 

To some extent, it is not surprising that the different cellular compartments exhibit coordinated transcriptional responses to WGD. The nucleus and organelles are in intimate communication via anterograde and retrograde signaling, and all known transcription factors regulating organelle transcription are encoded in the nucleus [2]. In fact, almost all proteins involved in organellar gene expression (RNA polymerases, transcription factors, ribosomal proteins) are encoded by the nucleus [2,37], so regulation of organellar gene expression is under direct control by the nucleus. In principle, therefore, changes to nuclear genome dosage should have a direct effect on organellar gene regulation.

Furthermore, the numbers of plastids and mitochondria per cell vary widely among tissue types and developmental stages [2,8,9,38], and even among cells of the same type [39] within diploids. This implies that cells are able to tolerate a wide range of nuclear/cytoplasmic genome dosages. This would, presumably, involve mechanisms to maintain expression homeostasis in response to changes in gene dosage.

Several lines of evidence suggest that doubling the nuclear genome would not cause significant disruption of cytonuclear interactions. First, organelles exist in dozens to hundreds of copies per cell, each with several copies of their genome [2]. Thus, doubling of nuclear genome copy number might have a relatively small effect on stoichiometry. For example, going from a state of having two nuclear genomes per 100 plastid genomes to a state of having four nuclear genomes per 100 plastid genomes is arguably a smaller perturbation (1:50 to 1:25) than if it involved going from 1:1 to 2:1 (as would be the case for a gene duplication affecting one of two nuclear genes encoding a heteromeric complex). 

Second, the number and/or volume of chloroplasts per cell often increases with WGD [35,40,41,42]. Increased plastid number per cell with WGD has been documented in maize, cotton, alfalfa (citations in [9]), and *Glycine* [43] among others [41]. The number of mitochondria per cell is also known to increase with cell size [44]. Given the correlation between WGD and cell size, therefore, the number of mitochondria per cell likely increases with ploidy as well [35]. 

Consistent with these observations, we find that the copy number per cell of both plastid- and mitochondrial genomes increased with WGD (Figure 1). In the absence of such an increase, the number of organellar genomes per nuclear genome would be expected to halve with WGD, yet tetraploids have approximately 75% as many plastid genomes per nuclear genome as their diploid progenitors (equating to approximately 1.5-fold as many plastid genomes per cell) and approximately equal numbers of mitochondrial genomes per nuclear genome (equating to twice as many mitochondrial genomes per cell). It is unclear from our qPCR-based assay if this reflects an increase in organelles per cell, genome number per organelle, or some combination of the two, but leaf cells respond to nuclear genome doubling by increasing organellar genome copies.

Consequently, any change in relative genome dosage produced by WGD is at least partially offset by increased plastid and mitochondrial DNA replication. On the organellar side, therefore, the maintenance of transcriptional balance we observe here may not, in fact, be due to transcriptional regulation per se, but due to an increase in organelle genome number. However, the extent to which organelle number increases with ploidy varies by species and tissue type [8,41], and consistent with our observations, Oberprieler et al. [8] found that plastid gene copy number did not keep pace with nuclear copy number in a *Leucanthemum* ploidy series. The fact that we still observe coordinated expression responses between the nucleus and plastid, therefore, indicates transcriptional regulation is involved in cytonuclear coordination, though perhaps only within the nucleus.

A limitation of the present study is that most regulation of organellar gene expression occurs post-transcriptionally [2]. As a consequence, transcript abundance and protein abundance do not exhibit strong correlations in plant mitochondria [45] or plastids. Thus, transcript abundances measured here give an incomplete picture of gene regulation in the organelles, and, therefore, of inter-genome coordination. It is possible, therefore that the nascent polyploids are experiencing dosage imbalance at the level of protein that would go undetected with transcriptomic data. If such imbalances are produced at the level of protein, this could explain the observation that nuclear genes whose products are organelle-targeted are among the first to return to singleton status after ancient polyploidy events [19]. A second potential limitation of the present study is the use of poly(A)-selected RNA for RNA-seq. Because polyadenylation typically targets plastid- and mitochondrial-transcripts for degradation [29,30], read counts from these compartments were low compared to similar libraries prepared with an rRNA-depletion strategy (Figure 5). Our normalized expression estimates for organellar genes correlated reasonably well with those from rRNA-depleted libraries (Figure 5), but it is likely that the accuracy and precision of our transcriptome profiles suffered to some extent from generally low read depths. Additional studies using rRNA-depleted RNA might, therefore, provide a clearer picture of transcriptional coordination between the different genomes. 

## 5. Conclusions

Our study provides evidence that diploid-like transcriptional balance between the nucleus, mitochondria and plastids is largely preserved in the first generations after nuclear genome duplication in *Arabidopsis thaliana*. This suggests that cells possess robust mechanisms to rapidly balance transcriptional outputs among the different genomes following WGD. One of these mechanisms is a compensatory increase in mitochondrial and plastid genome dosage, but additional layers of control at the level of transcription likely exist to maintain coordination between the different cellular compartments.

## Figures and Tables

**Figure 1 genes-11-00134-f001:**
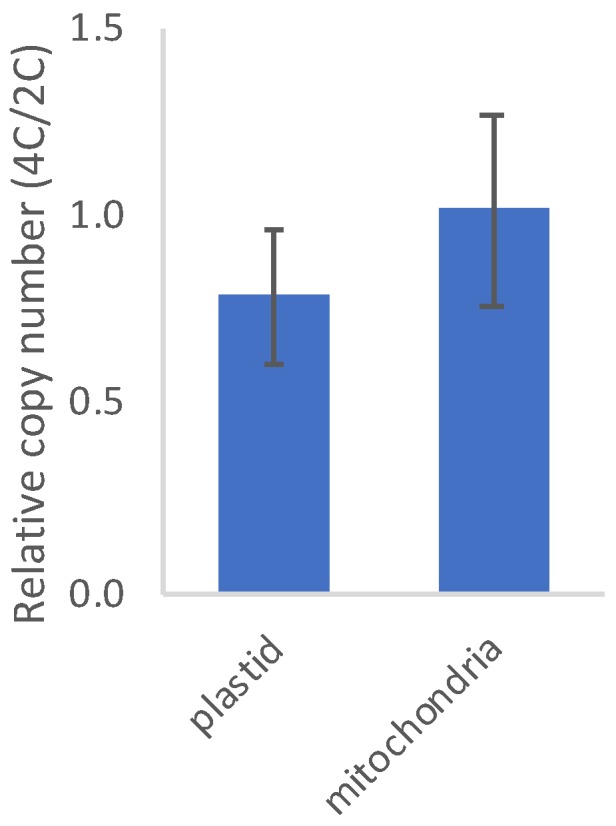
Quantitative PCR-based estimates of chloroplast- and mitochondrial-genome copy number relative to nuclear genome copy number in tetraploid vs. diploid rosette leaves.

**Figure 2 genes-11-00134-f002:**
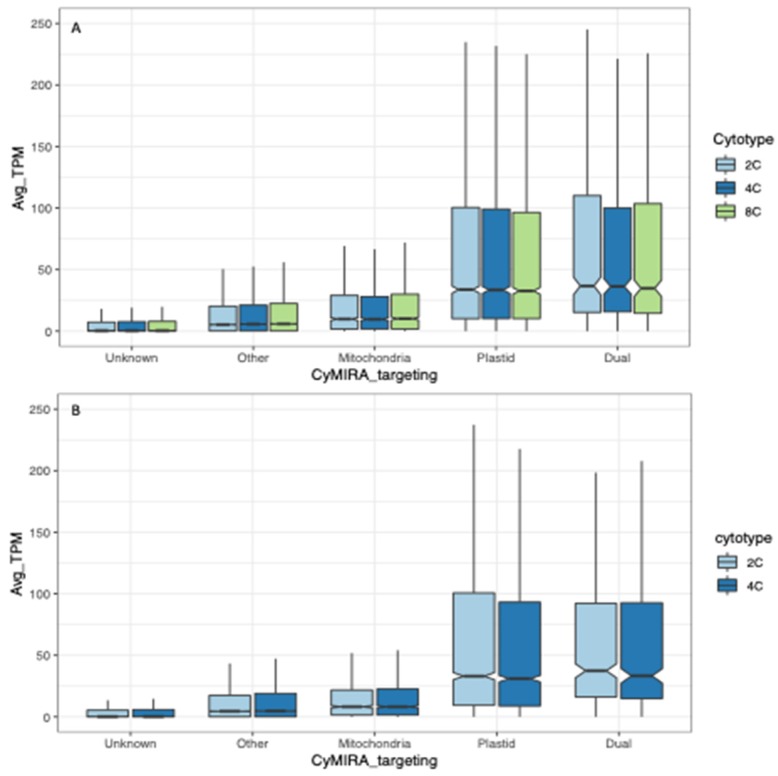
Expression level by subcellular target (obtained from the CyMIRA database) for nuclear genes in diploids (2C), tetraploids (4C) and octoploids (8C). (**A**) Sepals. (**B**) Leaves. For each cytotype, average transcripts per million (TPM) per locus was calculated for the 3–4 biological replicates per cytotype, and the distribution of these average values (Avg TPM; y-axis) was plotted by targeted compartment (x-axis).

**Figure 3 genes-11-00134-f003:**
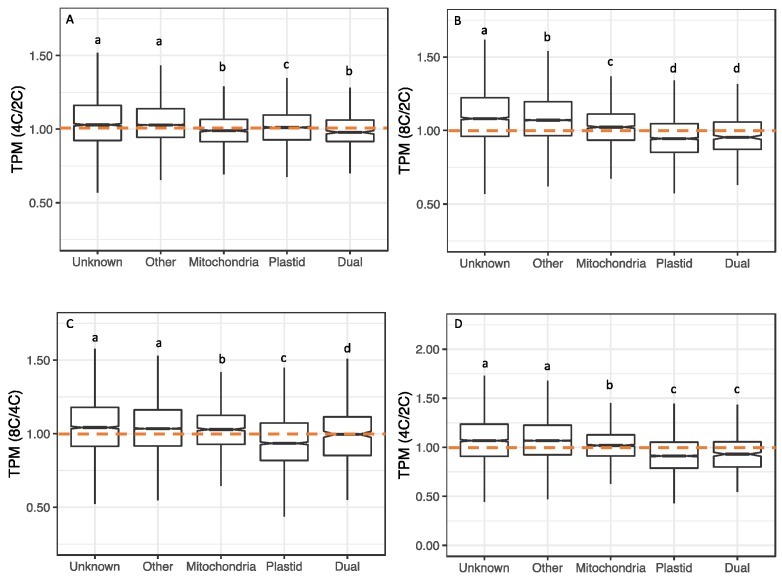
Polyploid vs. diploid expression (TPM) ratios by targeting class. (**A**) tetraploid (4C) vs. diploid (2C) sepals. (**B**) Octoploid (8C) vs. diploid sepals. (**C**) Octoploid vs. tetraploid sepals. (**D**) Tetraploid vs. diploid leaves. Differences in TPM ratios among targeting classes were assessed by Kruskal–Wallis tests with a post-hoc Dunn test (Benjamini–Hochberg adjustment). Different letters above boxes indicate significant differences (*p* < 0.05).

**Figure 4 genes-11-00134-f004:**
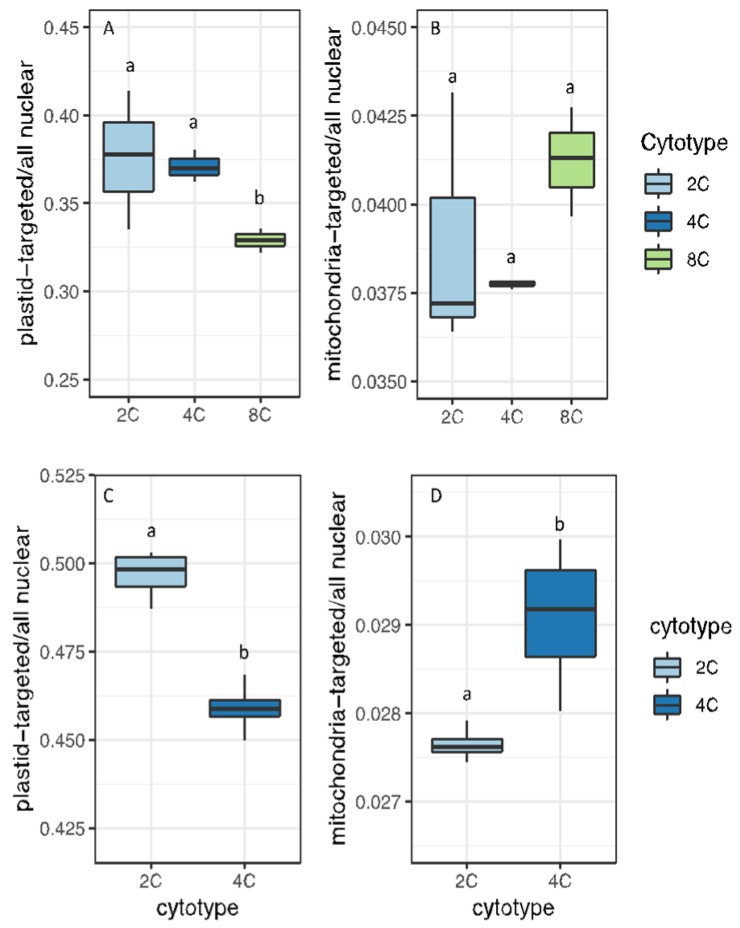
Ratios gene expression for nuclear genes with organelle-targeted products to total nuclear gene expression (read counts from genes with organelle-targeting divided by read counts of all nuclear genes). (**A**) Genes with plastid-targeting in sepals. (**B**) Genes with mitochondria-targeting in sepals. (**C**) Genes with plastid-targeting in leaves. (**D**) Genes with mitochondria-targeting in leaves. Different letters above boxes indicate significant differences (*p* < 0.05; t-tests with Bonferroni correction).

**Figure 5 genes-11-00134-f005:**
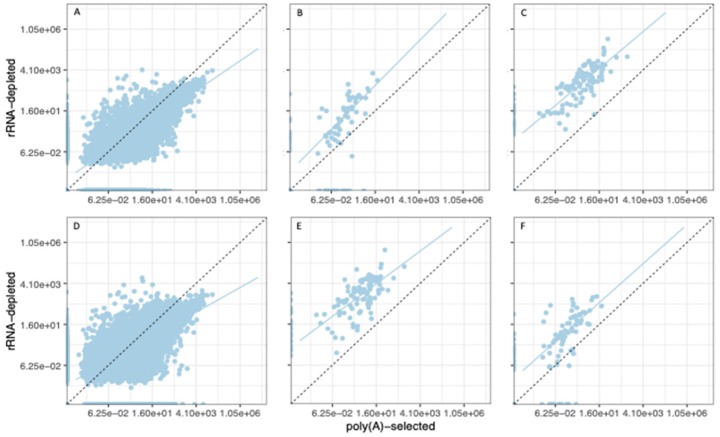
Scatterplots of expression estimates (TPM) from rRNA-depleted libraries vs. poly(A)-selected libraries. (**A**–**C**) Control libraries from SRA BioProject PRJNA380344. (**D**–**F**) Control libraries from SRA BioProject PRJNA437291. (**A**,**D**) Nuclear genes. (**B**,**E**) Plastid genes. (**C**,**F**) Mitochondrial genes. Dashed lines indicate unity (equal expression estimates). Solid lines are the line of best fit to the data.

**Figure 6 genes-11-00134-f006:**
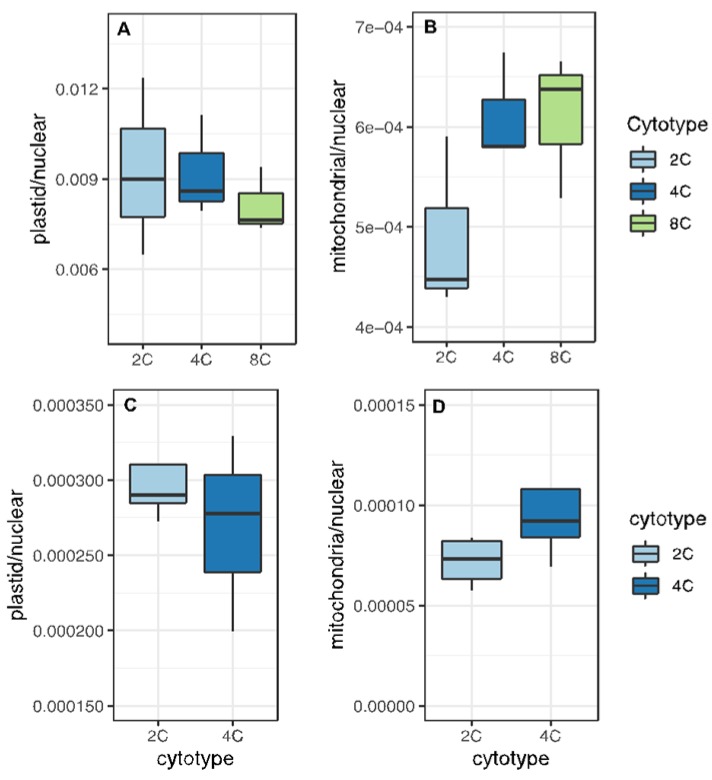
Ratios of organellar expression to nuclear expression. For each biological replicate per cytotype, total reads per organellar genome were divided by reads per nuclear genome. The distribution of organelle/nucleus ratios are shown. (**A**) Total plastid reads divided by total nuclear reads in sepals. (**B**) Total mitochondrial reads divided by total nuclear reads in sepals. (**C**) Total plastid reads divided by total nuclear reads in leaves. (**D**) Total mitochondrial reads divided by total nuclear reads in leaves.

**Figure 7 genes-11-00134-f007:**
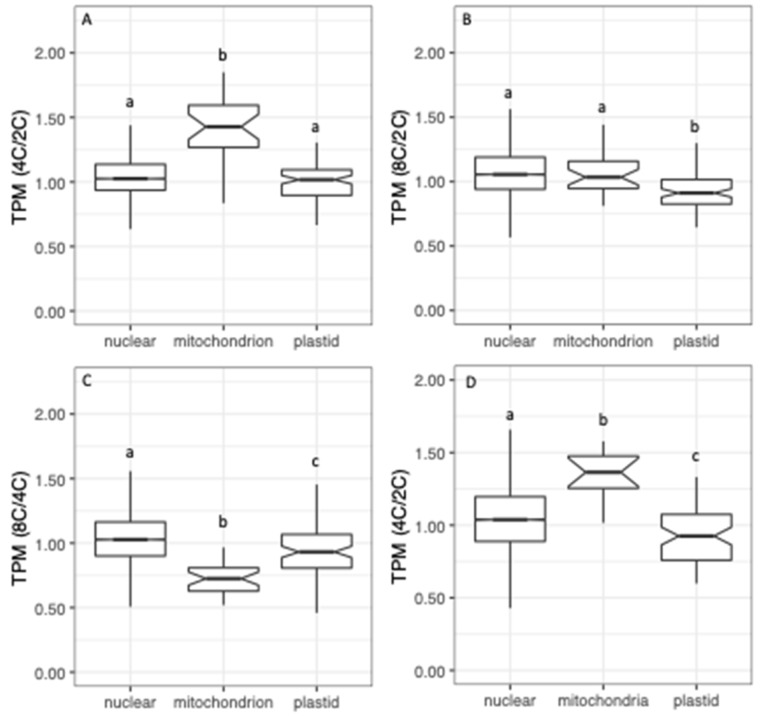
TPM ratios (polyploid/diploid) by genome. (**A**) Tetraploid vs. diploid sepals. (**B**) Octoploid vs. diploid sepals. (**C**) Octoploid vs. tetraploid sepals. (**D**) Tetraploid vs. diploid leaves.

**Figure 8 genes-11-00134-f008:**
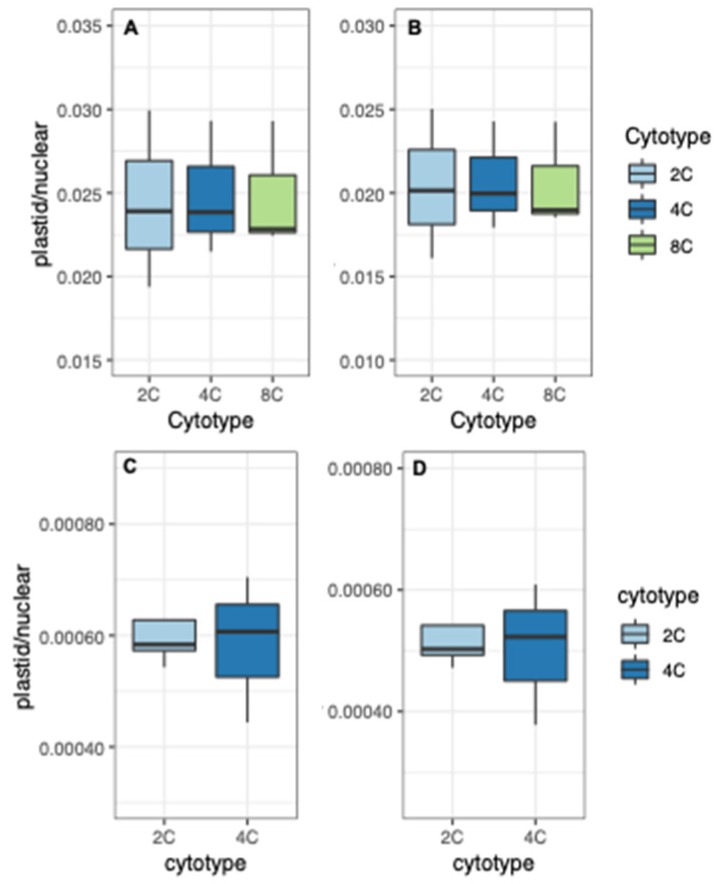
Ratios of plastid expression to plastid-targeted nuclear expression. (**A**) Total plastid reads divided by total plastid-targeted nuclear reads in sepals. (**B**) Total plastid reads divided by total plastid-targeted plus dual-targeted nuclear reads in sepals. (**C**) Total plastid reads divided by total plastid-targeted nuclear reads in leaves. (**D**) Total plastid reads divided by total plastid-targeted plus dual-targeted nuclear reads in leaves.

**Figure 9 genes-11-00134-f009:**
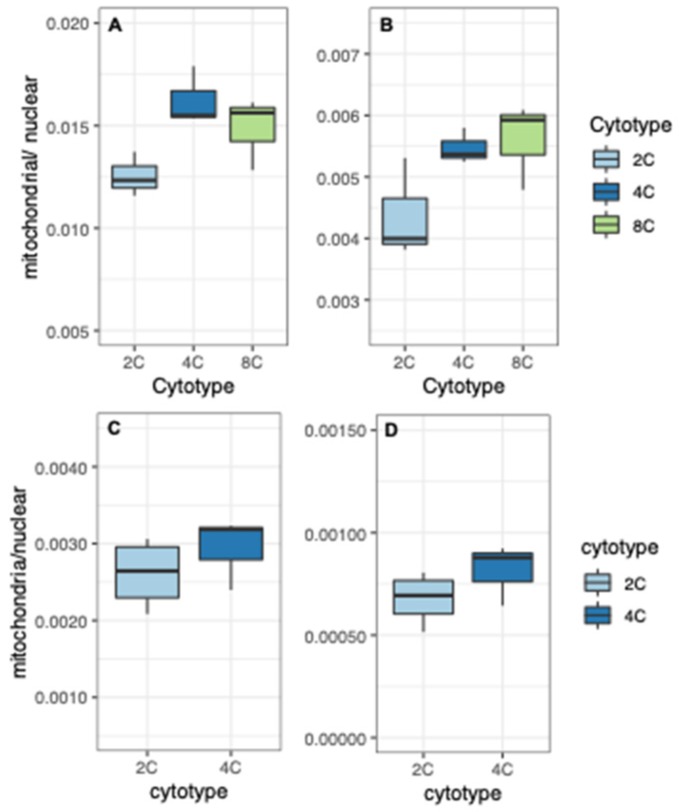
Ratios of mitochondrial expression to mitochondria-targeted nuclear expression. (**A**) Total mitochondrial reads divided by total mitochondria-targeted nuclear reads in sepals. (**B**) Total mitochondrial reads divided by total mitochondria-targeted plus dual-targeted nuclear reads in sepals. (**C**) Total mitochondrial reads divided by total mitochondria-targeted nuclear reads in leaves. (**D**) Total mitochondrial reads divided by total mitochondria-targeted plus dual-targeted nuclear reads in leaves.

**Figure 10 genes-11-00134-f010:**
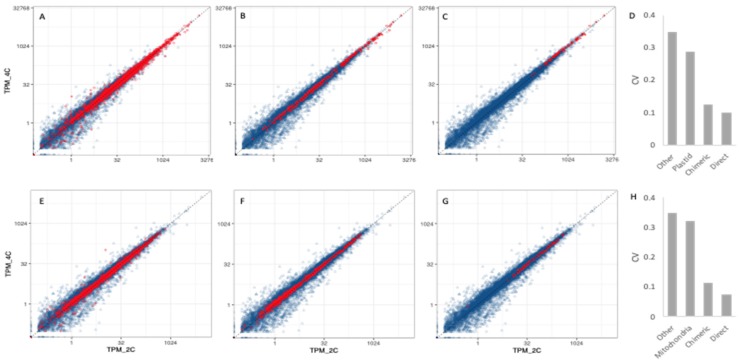
Expression (TPMs) in tetraploid sepals (4C; y-axis) vs. diploid sepals (2C; x-axis) for nuclear genes with targeting to organelles (red) or to other parts of the cell (blue). Blue data points are the same in all six panels (all genes with CyMIRA targeting of “other”, shown for comparison to the genes with organelle-targeting). Red data points represent (**A**) all genes with plastid-targeting; (**B**) genes whose proteins form chimeric complexes with plastid-derived DNA, RNA and/or proteins; (**C**) genes whose proteins have direct physical interactions with plastid-encoded proteins; (**D**) coefficients of variation (CV) of TPM ratios (4C/2C) for the different classes shown in panels **A**–**C**; (**E**) all genes with mitochondria-targeting; (**F**) genes whose proteins form chimeric complexes with mitochondria-derived DNA, RNA and/or proteins; (**G**) genes whose proteins have direct physical interactions with mitochondria-encoded proteins; (**H**) CV of TPM ratios (4C/2C) for the different classes shown in panels **E**–**G**.

**Figure 11 genes-11-00134-f011:**
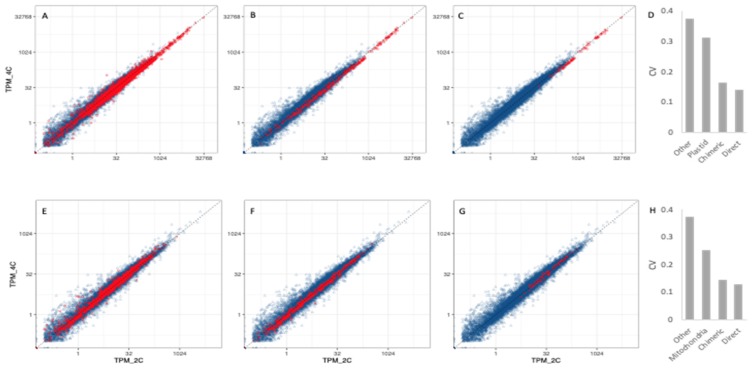
Expression (TPMs) in tetraploid leaves (4C; y-axis) vs. diploid leaves (2C; x-axis) for nuclear genes with targeting to organelles (red) or to other parts of the cell (blue). Blue data points are the same in all six panels (all genes with CyMIRA targeting of “other”, shown for comparison to the genes with organelle-targeting). Red data points represent (**A**) all genes with plastid-targeting; (**B**) genes whose proteins form chimeric complexes with plastid-derived DNA, RNA and/or proteins; (**C**) genes whose proteins have direct physical interactions with plastid-encoded proteins; (**D**) coefficients of variation (CV) of TPM ratios (4C/2C) for the different classes shown in panels **A**–**C**; (**E**) all mitochondria-targeted genes; (**F**) mitochondria-targeted genes whose proteins form chimeric complexes with mitochondria-derived DNA, RNA and/or proteins; (**G**) mitochondria-targeted genes whose proteins have direct physical interactions with mitochondria-encoded proteins; (**H**) CV of TPM ratios (4C/2C) for the different classes shown in panels **E**–**G**.

**Figure 12 genes-11-00134-f012:**
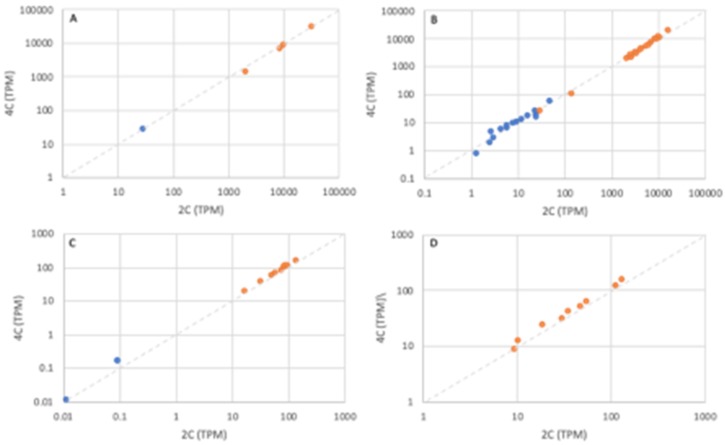
Expression (TPMs) in tetraploid leaves (4C; y-axis) vs. diploid leaves (2C; x-axis) for selected organellar protein complexes. (**A**) RuBisCO, (**B**) Photosystem II. (**C**) OXPHOS complex III, (**D**) OXPHOS complex II (all subunits encoded in the nucleus). Nuclear-encoded subunits are shown in orange, plastid- and mitochondria-encoded subunits in blue.

**Figure 13 genes-11-00134-f013:**
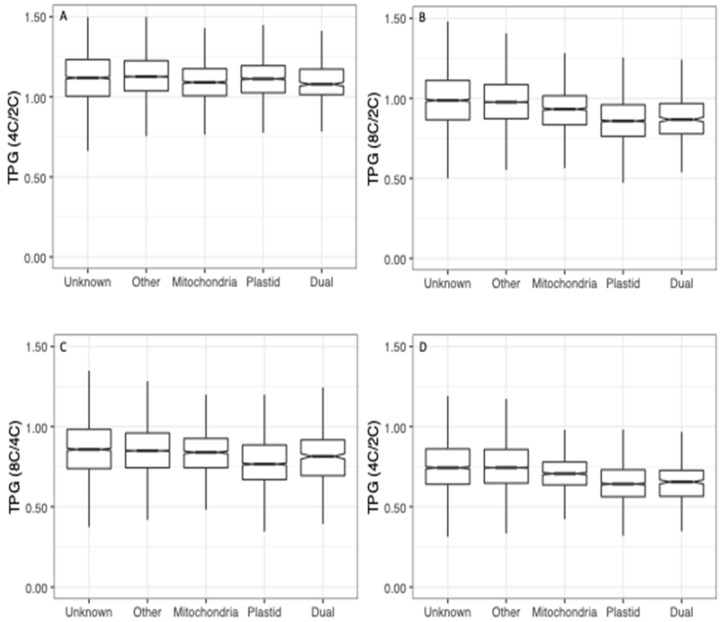
Transcripts per genome for nuclear genes by targeting class. (**A**) Tetraploid vs. diploid (4C/2C) in sepals. (**B**) Octoploid vs. diploid (8C/2C0 in sepals. (**C**) Octoploid vs. tetraploid (8C/4C) in sepals. (**D**) Tetraploid vs. diploid (4C/2C) in leaves.

**Table 1 genes-11-00134-t001:** Linear regressions of average TPM: poly(A)-selected libraries vs. rRNA-depleted libraries (control treatments specified Bioprojects). F = F-statistic; DF = degrees of freedom; *p* = *p*-value; r = correlation coefficient.

Comparison	Genome	F	DF	*p*	R
Poly(A) vs. PRJNA380344	Mitochondrion	140.9	1, 150	<0.0001	0.696
	Plastid	4.242	1, 131	0.041	0.177
	Nuclear	3.1 × 10^4^	1, 32907	<0.0001	0.696
Poly(A) vs. PRJNA437291	Mitochondrion	77.86	1, 150	<0.0001	0.585
	Plastid	4.02	1, 131	0.047	0.172
	Nuclear	1.8 × 10^3^	1, 32907	<0.0001	0.229

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
