# Peer review of "Robust Cytonuclear Coordination of Transcription in Nascent Arabidopsis thaliana Autopolyploids"

_genes, 2020, doi:10.3390/genes11020134_

Round 1
Reviewer 1 Report
In this elegantly crafted manuscript data are presented bearing on the question of coordination between organellar and nuclear genomes in response to polyploidy. The study system selected was diploid vs. autopolyploid Arabidopsis thaliana, using a combination of tools designed to quantify transcriptomic responses to altered genome copy numbers. All of the results are significant, original, and are supported by the data presented. They will be well-received by the large community interested in whole genome doubling in plants, and others interested in cytonuclear regulation. I have no criticisms of the ms, and only note the tiniest of typos or wording matters which the authors might wish to address. These are below; for now, and in summary, bravo!
Line 15: maybe should be past tense (increased, not increases)
Lines 35-36: "has been the major focus of a rather limited body" seems a bit awkward... simplify?
Lines 180-185: it might be helpful to add a bit of discussion or perspective on the precision and accuracy of choosing just one gene each for organellar estimation, and two genes for nuclear estimation
Line 209 and elsewhere: genes are not targeted to plastids, of course, but their protein products are; it would be good to check throughout for this bit of linquistic precision
Line 214: something wrong with legend....C should be leaves, right?
Lines 230-231: meaning is clear, but "investment" seems to be a bit of an overreach.
Lines 297-299: some redundancy in the figure legend
Lines 338-339: same issue with genes targeted....
Line 396: Song et al, in revision... I didn't see it in the literature cited
Author Response
We thank the editors and three expert reviewers for their helpful feedback on our manuscript. Based on these comments, we have revised the manuscript with all changes tracked in the attached Word document (genes-698208_revised011720.docx), and the changes are detailed below.
[reviewer comments are in plain text, our responses in bold]
[for revised text, deleted text is shown in strikethrough and new text is underlined]
Reviewer 1
In this elegantly crafted manuscript data are presented bearing on the question of coordination between organellar and nuclear genomes in response to polyploidy. The study system selected was diploid vs. autopolyploid Arabidopsis thaliana, using a combination of tools designed to quantify transcriptomic responses to altered genome copy numbers. All of the results are significant, original, and are supported by the data presented. They will be well-received by the large community interested in whole genome doubling in plants, and others interested in cytonuclear regulation. I have no criticisms of the ms, and only note the tiniest of typos or wording matters which the authors might wish to address. These are below; for now, and in summary, bravo!
We thank the reviewer for their kind assessment of the manuscript!
Line 15: maybe should be past tense (increased, not increases)
Suggestion accepted.
Lines 35-36: "has been the major focus of a rather limited body" seems a bit awkward... simplify?
We have deleted “of a rather limited body” from this sentence.
Lines 180-185: it might be helpful to add a bit of discussion or perspective on the precision and accuracy of choosing just one gene each for organellar estimation, and two genes for nuclear estimation
This is a valid point, and we have added the following text to address it, and qualify these observations:
Line 244: “The qPCR data indicate that plastid copy number decreased slightly per nuclear genome (0.76-fold +/- 0.18 SE) in tetraploids vs. diploids, whereas mitochondrial copy number remained constant (1.01-fold +/- 0.25 SE) (Fig. 1).”
Line 246: “Thus, tetraploids appear to compensated for increased nuclear genome dosage with…”
Lines 248-252: Added “Note, however, that because the relative copy numbers reported here are based on limited gene sampling (two nuclear genes and two organellar genes), the estimates should be considered approximate. Quantitative PCR with additional genes and/or direct counting by microscopy will be necessary to validate and refine these estimates.”
Line 209 and elsewhere: genes are not targeted to plastids, of course, but their protein products are; it would be good to check throughout for this bit of linquistic precision
We have revised throughout. This involved MANY changes, so not all are listed here, but a few examples include:
Line 18: “plastid-targeted nuclear genes whose products are plastid-targeted…”
Line 20: “mitochondria-targeted nuclear genes with mitochondrial targeting…”
Line 67: “Organelle-targeted nNuclear genes whose products are targeted to organelles…”
Line 279-282: “In both tissues organs, the expression of genes with whose products have unknown or non-organelle targeting (CyMIRA targeting designation of “other”) increased with WGD (e.g., 4C/2C ratio >1) whereas the expression of genes whose products are targeted to plastids…”
Line 214: something wrong with legend....C should be leaves, right?
The confusion is understandable, but because we have three ploidy levels for sepal tissue the first three panels (A-C) are in fact all for sepals (A=4C/2C, B=8C/2C, and C=8C/4C).
Lines 230-231: meaning is clear, but "investment" seems to be a bit of an overreach.
Revised to: “Consequently, as ploidy increases, relative transcriptional output within the nucleus increases for genes with mitochondrial targeting and decreases for genes with plastid targeting.”
Lines 297-299: some redundancy in the figure legend
Thank you for catching this. We have deleted the inappropriate text (“B) Octoploid vs. diploid leaves. C) Octoploid vs. tetraploid leaves.”
Lines 338-339: same issue with genes targeted....
Revised as follows:
Line 482-483: “…genes targeted with targeting to both organelles…”
Line 483: “…genes whose products are targeted…”
Lines 486: changed “than organelle-targeted genes” to “than do genes with organelle-targeting”
Line 396: Song et al, in revision... I didn't see it in the literature cited
Thank you for catching this. The reference has been added:
Song, Michael, J., Potter, B., Doyle, J.J., and Coate, J.E. (in revision). Gene balance predicts transcriptional responses immediately following ploidy change in Arabidopsis thaliana. Plant Cell.
Reviewer 2 Report
Here the authors evaluate the response of organellar transcriptomes (plastid and mitochondria) to increasing ploidy. This response is vastly understudied, and the reconciliation of potential stoichiometric imbalances is still largely unknown.
Overall, I really like this paper. It is well written and clearly laid out. I have two significant comments. The first is relatively minor; the statistics of the analyses are not listed in the methods. How were they conducted? What was the design? What was the software?
The second concern I waffle on. The authors use poly-A selected transcripts to represent expression in the organellar genomes. I imagine this was done to reuse data (while presented first here, perhaps was designed for a different project?), and they do analyze other data (also previously generated) to suggest that the transcription profiles of the poly-A generated libraries correlate with rRNA-depleted libraries, which is preferred for organellar expression. I waffle because their analysis does seem to suggest that the poly-A expression profiles do more-or-less represent what you would find in rRNA-depleted libraries; however, research generally suggests that the organelles retain the prokaryotic state whereby polyadenylation of these transcripts (in plants) is used to mark the transcript for degradation. (See Lange et al, 2009, Trends in Plant Science and Schuster and Stern, 2009, Progress in Molecular Biology and Translational Science) I do like that the authors address this with data; however, perhaps they can speak to the reconciliation between these prior observations and their finding of correlation. Where may some pitfalls lie?
In general, however, I think the authors did a nice job.
Author Response
We thank the editors and three expert reviewers for their helpful feedback on our manuscript. Based on these comments, we have revised the manuscript with all changes tracked in the attached Word document (genes-698208_revised011720.docx), and the changes are detailed below.
[reviewer comments are in plain text, our responses in bold]
[for revised text, deleted text is shown in strikethrough and new text is underlined]
Here the authors evaluate the response of organellar transcriptomes (plastid and mitochondria) to increasing ploidy. This response is vastly understudied, and the reconciliation of potential stoichiometric imbalances is still largely unknown.
Overall, I really like this paper. It is well written and clearly laid out.
Thank you!
I have two significant comments. The first is relatively minor; the statistics of the analyses are not listed in the methods. How were they conducted? What was the design? What was the software?
We apologize for this omission, and have added the following:
Methods (Lines 200-214):
“Quantifying transcription: Read counts per gene were determined from the BAM files using HTSeq [25] with the “intersection non-empty” setting using the Araport11 annotation file (Athaliana_447_Araport11.gene_exons.gff3) downloaded from Phytozome (https://phytozome.jgi.doe.gov/pz/portal.html). Normalized read counts (Transcripts Per Million [TPM]) were calculated by first calculating Reads Per Kilobase (RPK; read count per gene divided by gene length in Kb), summing RPK values for all genes, then dividing individual RPK values by the sum of all RPK values divided by 1 million.
Statistics: TPM ratios (e.g., 4C/2C) were compared among targeting classes or genomes using Kruskal-Wallis tests, followed by Dunn’s multiple comparisons tests with Benjamini-Hochberg adjustments for multiple comparisons. Total read count ratios were compared by one-way ANOVA followed by t-tests with Bonferroni correction for multiple comparisons. Correlations between expression estimates from poly(A)-selected libraries and ribosomal RNA-depleted libraries (see below) were assessed by one-way ANOVA. All statistical tests were implemented in R. Dunn tests were implemented using the FSA package.”
Fig. 3 legend (Lines 289-293): Added “Differences in TPM ratios among targeting classes were assessed by Kruskal-Wallis tests with a post-hoc Dunn test (Benjamini-Hochberg adjustment). Different letters above boxes indicate significant differences (p < 0.05).”
The second concern I waffle on. The authors use poly-A selected transcripts to represent expression in the organellar genomes. I imagine this was done to reuse data (while presented first here, perhaps was designed for a different project?), and they do analyze other data (also previously generated) to suggest that the transcription profiles of the poly-A generated libraries correlate with rRNA-depleted libraries, which is preferred for organellar expression. I waffle because their analysis does seem to suggest that the poly-A expression profiles do more-or-less represent what you would find in rRNA-depleted libraries; however, research generally suggests that the organelles retain the prokaryotic state whereby polyadenylation of these transcripts (in plants) is used to mark the transcript for degradation. (See Lange et al, 2009, Trends in Plant Science and Schuster and Stern, 2009, Progress in Molecular Biology and Translational Science) I do like that the authors address this with data; however, perhaps they can speak to the reconciliation between these prior observations and their finding of correlation. Where may some pitfalls lie?
The reviewer is correct in the inference that these data were generated using poly(A) selection because we did not originally intend to assess plastid or mitochondrial expression. The reviewer also raises a valid concern – that poly(A) transcripts in the organelles are usually targeted for degradation (which is, indeed why we compared our data to rRNA-depleted libraries).
We added the references provided by the reviewer (Lange at al. and Schuster and Stern) and the following text (Lines 368-369): “…transcripts from organellar protein-coding genes are not always polyadenylated [27], and generally targeted for degradation if they are [28,29].”
We also added the following text to the Discussion (lines 696-703) to address this potential pitfall: “A second potential limitation of the present study is the use of poly(A)-selected RNA for RNA-seq. Because polyadenylation typically targets plastid- and mitochondrial-transcripts for degradation [28,29], read counts from these compartments were low compared to similar libraries prepared with an rRNA-depletion strategy (Fig. 5). Our normalized expression estimates for organellar genes correlated reasonably well with those from rRN-depleted libraries (Fig. 5), but it is likely that the accuracy and precision of our transcriptome profiles suffered to some extent from generally low read depths. Additional studies using rRNA-depleted RNA might, therefore, provide a clearer picture of transcriptional coordination between the different genomes.”
In general, however, I think the authors did a nice job and I find this manuscript to be suitable for publication in Genes.
Thank you!
Reviewer 3 Report
Authors,
Very nice paper covering an important aspect of plant evolution - How genomic compartments buffer against changes in gene dosage after an autopolyploid WGD event. I hope to read more papers from the authors in future on this and related topics.
Please consider the following things in your revision:
1) Please use the term tissue very carefully. Generally when one is doing DNA genotyping the term tissue refers to any part of the plant used to acquire DNA. However since you are doing a transcript comparison study you must specify that you are not doing tissue specific comparisons as is often done to see how a given biosynthetic pathway is partitioned between different tissues in a plant organ such as comparisons between transcriptional activity in the epidermis and vascular tissue of the leaf organ. I suggest using the terms rosette leaf and sepal throughout the manuscript. I have included several comments in the text of the document in regards to this subject.
2)Please make very clear in the intro the differences between auto- and alloployploid WGD as well as the different expectations for genomic reorganization after these two events - and that your paper will be looking at autopolyploidy.
3)Avoid referential sentences - that is do not simply refer to something in the previous sentence with 'it' or 'these' but rather state the thing that you want to compare to or reference.
4)Numerous other edits are included in the attached document for your consideration.

Author Response
We thank the editors and three expert reviewers for their helpful feedback on our manuscript. Based on these comments, we have revised the manuscript with all changes tracked in the attached Word document (genes-698208_revised011720.docx), and the changes are detailed below.
[reviewer comments are in plain text, our responses in bold]
[for revised text, deleted text is shown in strikethrough and new text is underlined]
Very nice paper covering an important aspect of plant evolution - How genomic compartments buffer against changes in gene dosage after an autopolyploid WGD event. I hope to read more papers from the authors in future on this and related topics.
Thank you!
Please consider the following things in your revision:
1) Please use the term tissue very carefully. Generally when one is doing DNA genotyping the term tissue refers to any part of the plant used to acquire DNA. However since you are doing a transcript comparison study you must specify that you are not doing tissue specific comparisons as is often done to see how a given biosynthetic pathway is partitioned between different tissues in a plant organ such as comparisons between transcriptional activity in the epidermis and vascular tissue of the leaf organ. I suggest using the terms rosette leaf and sepal throughout the manuscript. I have included several comments in the text of the document in regards to this subject.
We thank the reviewer for this suggestion and have changed the wording throughout, primarily by replacing “tissue” with “organ”. For example:
Line 13 and line 76: Change “…different tissue types (sepals and leaves)” to “…different organs (sepals and rosette leaves)”
Line 142: Changed “Leaf” to “Mature rosette leaf”
Fig. 1 legend: Changed “leaf tissue” to “rosette leaves”
Line 260: Changed “tissues” to “organs” and “leaves” to “rosette leaves”
2)Please make very clear in the intro the differences between auto- and alloployploid WGD as well as the different expectations for genomic reorganization after these two events - and that your paper will be looking at autopolyploidy.
We attempted to do this with the following additions:
Lines 34-37: Revised as follows: “Allopolyploidy, which involves the merger of two or more differentiated nuclear genomes, also brings together novel combinations of nuclear and cytoplasmic genomes, potentially creating cytonuclear incompatibilities,. This has been the major focus of research on cytonuclear interactions in polyploids [9,12–16].”
Lines 39-66: Revised to: “Because many protein complexes in organelles consist of both organelle-encoded subunits and nuclear-encoded subunits [17], altering relative gene dosage potentially disrupts the assembly and/or function of these chimeric complexes [18]. This could have immediate deleterious consequences in both allo- and autopolyploids (polyploids resulting from genome multiplication in the absence of hybridization). This aspect of cytonuclear interactions has received less attention than the cytonuclear incompatibilities specifically associated with allopolyploidy [8].”
Lines 97-100: Added “By utilizing nascent autopolyploids, we could assess the immediate effects of nuclear genome multiplication, independent of hybridization, on transcriptional coordination between the different genome-containing compartments of the cell.”
3)Avoid referential sentences - that is do not simply refer to something in the previous sentence with 'it' or 'these' but rather state the thing that you want to compare to or reference.
We searched for instances of referential statements, and fixed the following:
Line 278-279: “These dDistributions of TPM ratios differed significantly by targeting category…”
Line 316: “To see if these patterns this restructuring of the nuclear transcriptome reflects coordination with organellar transcription, 4)Numerous other edits are included in the attached document for your consideration.”
Lines 370: “Thus, it is possible that read counts from these organellar genes may not reflect the relative abundances of the transcripts produced.”
Lines 376-377: “We then compared normalized expression estimates (TPMs) from these rRNA-depleted libraries to those of our diploid libraries.”
Line 399: “…and that these poly(A)-selected libraries, therefore, accurately report relative expression…”
Fig. 6 legend: “The distribution of these three organelle/nucleus ratios are shown.”
Line 557: “These Such relative measures report…”
Line 633: Replaced “here” with “in the case of mitochondria”
4) Numerous other edits are included in the attached document for your consideration.
We thank the reviewer for their careful reading of the manuscript and suggested edits. We have accepted most of these as follows.
Title: Added “auto” to “polyploids”. Also, though his was not suggested by the reviewers, we revised “Robust transcriptional cytonuclear coordination” to “Robust cytonuclear coordination of transcription” because it seems to flow better. Revised title is now: “Robust cytonuclear coordination of transcription in nascent Arabidopsis thaliana autopolyploids”
Line 11: Added “auto”
Line 30: Changed “co-evolve” to “co-evolved”
Line 31: Changed “organismal fitness” to “cellular function”
Line 33: Deleted “Managing”
Line 36: Replace “This” with “These incompatibilities”
Line 42: Added citation [18] = Birchler et al. 2016
Lines 44-45: Deleted “potentially” and added “resulting”
Line 76: Added “species of” (the reviewer asked which species of Leucanthemum were studied – the added text is intended to make it clear that several species were involved)
Line 83: Replaced “meaning” with “indicating that”
Line 90: Deleted “pure”
Line 96: Added taxonomic authority, “(L.) Heynh”
Line 136: Changed “Sunshine #4 potting mix” to “Mix #4 growing media (Sunshine ®)”
Lines 166-169: Added “Different library protocols were used for sepals and leaves because we generated leaf libraries in house, whereas sepal libraries were made at a core facility. Because all comparisions were between ploidy levels within an organ (leaves or sepals), the different library protocols should not affect interpretation of results.”
Lines 233-234: Added “Libraries were selected from two different experiments to have independent assessments of the correlations between poly(A)-selected and rRNA-depleted libraries.”
Line 277: Changed “(e.g., tetraploid/diploid, octoploid/tetraploid)” to “(tetraploid/diploid, octoploid/diploid, and octoploid/tetraploid)”
Fig. 3: Labels were added to panels C and D
Line 295: Changed “conclusion” to “finding”
Table 1 legend: Added “F = F-statistic; DF = degrees of freedom; p = p-value; r = correlation coefficient.”
Line 417: Changed “Again, we see a substantially similar pattern…” to “We see a similar pattern…”
Line 460: Deleted “per cell”
Line 477: Replaced “direct physical contact” with “direct interaction”
Line 541: Replaced “looked at” with “analyzed”
Line 610: Added “auto” to “Autopolyploids”
Line 623: Replaced “they” with “plastid genes”
Line 633: Replaced “here” with “in the case of mitochondria”
Line 635: Replaced “resist” with “buffer”
Line 636: Replaced “In one sense” with “To some extent”
Lines 642: Deleted “knock-on”
Line 649: Deleted “Additionally”
Reviewer commented, regarding the possibility of polyploids increasing organelle number, that we should “make this more evident – i.e., say something about it in the intro”. Consequently, we the sentence on lines 70-73 to: “Does nuclear genome doubling alter the stoichiometry of nuclear-encoded vs. organelle-encoded subunits of chimeric protein complexes, or is inter-genome coordination capable of compensating for nuclear genome doubling (e.g., via increasing the number of organelles and/or organellar transcription)?”